# The effects of selected biologics and a small molecule on Health-Related Quality of Life in adult plaque psoriasis patients: A systematic review and meta-analysis

Anna Karpińska-Mirecka[ID]*, Joanna Bartosińska[¤], Dorota Krasowska[ID]

Department of Dermatology, Venereology and Pediatric Dermatology, Medical University of Lublin, Lublin, Poland

¤ Current address: Cosmetology and Aesthetic Medicine Unit, Medical University of Lublin, Lublin, Poland

* karpinska@interia.eu

**Data Availability Statement:** All relevant data are within the paper and its Supporting Information files.

## Abstract

### Background

The Dermatology Life Quality Index (DLQI) is commonly used to assess the quality of life of patients with skin diseases. Clinical trials confirm the positive effect of the use of biologics and new molecules on the quality of life of patients with plaque psoriasis.

### Main objectives

Investigation of the effect of infliximab, adalimumab, ixekizumab, secukinumab and tofacitinib on Health-Related Quality of Life (HRQOL) measured by the DLQI in adult plaque psoriatic patients with respect to the patients' race, type of used agent/placebo, agent's dosage and treatment duration as well as the DLQI score prior to and after commencement of treatment.

### Material and methods

Systematic literature searching for referential papers written in English using four databases: PubMed, EMBASE, Scopus, ClinicalTrials.gov as well as ~~and~~ manual searching (Google) Cochran's (Q) and $I^2$ tests were used for evaluation of heterogeneity or the degree of variation in the true effect size estimates between the analysed studies. The standardized mean difference (the SMD; Hedge's g score) was applied to measure the differences between the two means (i.e. two groups: treated vs non-treated or treated vs placebo). The data coding and Hedge's g values were calculated according to the guidance of MetaXL software version 5.3.

### Main results

43 studies, in total 25,898 individuals, were evaluated by the DLQI and weighted mean scores were derived for the analysis. The mean DLQI scores ranged from 6.83 to 17.8 with the overall DLQI score of 12.12 (95%CI: 11.24 to 13.06). A random-effects model demonstrated significant considerable heterogeneity of the study results ($I^2$ = 98%; p<0.001).

**Funding:** The authors received no specific funding for this work.

**Competing interests:** The authors have declared that no competing interests exist.

## Conclusion

Infliximab, adalimumab, ixekizumab, secukinumab and tofacitinib in adult plaque psoriatic patients improved HRQOL measured by the DLQI. The patients with lower quality of life before treatment obtained better results.

## Introduction

Psoriasis, an incurable chronic inflammatory disease affecting approximately 2% of people worldwide, with the lowest incidence in the Asian and some African populations as well as the highest rate, i.e. up to 11%, in the Caucasian and Scandinavian populations, is most commonly observed in the form of plaque psoriasis (in 80% of all diagnosed cases) [1, 2]. It is often accompanied by various comorbidities, such as psoriatic arthritis, Crohn's disease, metabolic syndrome, cardiovascular diseases, all of which account for the psoriatic patients' compromised quality of life frequently manifested by social withdrawal and hampered daily activities [3, 4]. This psychological aspect of psoriasis has found a reflection in the recommendations of the European Consensus on psoriasis treatment goals [5]. What is more, the use of biological drugs and new molecules appears to be improving the quality of life of the patients suffering from this debilitating disease [6]. The choice of treatment, however, depends on its severity, localization of the skin symptoms and the patient's preferences and needs [5, 7].

In order to measure the severity of psoriasis, the Psoriasis Area and Severity Index (PASI), Body Surface Area (BSA) and the Dermatology Life Quality Index (DLQI) are used. The DLQI, a 10-item questionnaire, covers six domains of daily life, such as symptoms and feelings, daily activities, leisure, work and school, personal relationships and treatment. It is used in daily clinical practice as well as in clinical trials and its score ranges from 0 to 30. Thus, according to the European Consensus, BSA$\geq$10 or PASI $\geq$10 or DLQI $\geq$ 10 qualify for the systemic treatment [5].

The first biologics used in the systemic treatment of psoriasis were the tumor necrosis factor (TNF) inhibitors (infliximab, etanercept, adalimumab, golimumab, certolizumab pegol) [6, 8]. Further development of biologics involved introduction of other biological drugs, i.e. the monoclonal antibodies that block interleukin 12 and 23 (ustekinumab) or target interleukin 23 (guselkumab, tildrakizumab, risankizumab) as well as those which are capable of blocking interleukin 17 (secukinumab, ixekizumab and brodalumab) [9–11]. It is worth noting that tofacitinib, which is one of the synthetic small molecules, plays an important role in the psoriasis treatment [12].

The DLQI is used to check for any correlations between the used biologics and improved quality of life in adult plaque psoriatic patients. Therefore, the main objective of the study was to investigate the effects of infliximab, adalimumab, ixekizumab, secukinumab and tofacitinib on Health-Related Quality of Life measured by the DLQI in adult plaque psoriatic patients taking into account the patients' race, the type of used agent/placebo, the agent's dosage and treatment duration as well as the DLQI score prior to and after the commencement of the treatment.

## Methods

### Search strategy

The study was designed in accordance with the Preferred Reporting Items for Systematic Reviews and Meta-Analyses (PRISMA) guidelines [13]. We conducted systematic literature

searching using four databases: PubMed (until October, 2019), EMBASE (until October, 2019), Scopus (until October, 2019), ClinicalTrials.gov and manual searching (Google) for papers written in the English language. The literature search was limited to the years from 2000 to the end of October 2019.

The following approach of literature searching was used with the application of keywords: "psoriasis", "quality of life", "life quality", "life", "standard of living", "living standards", "Dermatology Life Quality Index", "health-related quality of life", then supplemented by "plaque", "chronic", "severe", "DLQI" and "HRQOL". Also, the following search builder was used: ("plaque psoriasis" OR "psoriasis") AND ("quality of life" OR "life quality" OR "standard of living" OR "living standards" OR "health-related quality of life") Additionally, the PubMed database was searched with the use of MeSH terms, as follows: (Psoriasis and Quality of life) "Psoriasis" [MeSH Terms] AND ("Quality of life" [MeSH Terms] OR "Quality of life" [All fields]) AND ("Socioeconomic Factors" [MeSH Terms] OR ("Socioeconomic Factors" [All fields]).

Our research was based on the PRISMA guidelines derived from http://www.prisma-statement.org/ and https://www.ncbi.nlm.nih.gov/pmc/articles/PMC6461330/.

## Eligibility criteria

For the systematic review, both inclusion and exclusion criteria were selected. The former were as follows: 1. Study enrollment of adult patients, 2. Study enrollment of patients with plaque psoriasis, 3. Ability to extract the DLQI scores, 4. A sample group of ten or more. The latter criteria were as follows: 1. A review/case report study type, 2. Language other than English, 3. Other than plaque psoriasis clinical types of the disease, 4. Studies with no QoL assessment according to the DLQI. Eventually, we went for: the principal author, publication year, the number of studied patients, treatment regimen and its duration as well as the DLQI score prior to and after the treatment.

## Data analysis

All the steps of meta-analysis were conducted with the use of the MetaXL software version 5.3 (EpiGear). Heterogeneity or the degree of variation in the true effect size estimates between the studies was evaluated using Cochran's (Q) and $I^2$ tests. The $I^2$ test was used to assess the heterogeneity among the analysed studies—if the $I^2$ value was between 0–40%, heterogeneity was rejected, its range between >40%-70% indicated substantial heterogeneity, while the scores >75% represented considerable heterogeneity. A random-effects model was preferred for heterogeneity presentation. The standardized mean difference (the SMD; Hedge's g score) was applied to measure the differences between the two means (i.e. two groups: treated vs non-treated or treated vs placebo). Hedge's g over 0.8 indicates a large effect size. A funnel plot graph, Doi plots and the Luis Furuya-Kanamori (LFK) index methods were used to check the existence of publication bias. LFK <1 indicates no asymmetry, LFK between 1–2 considers minor asymmetry, while LFK>2 suggests major asymmetry [14]. p values below 0.05 were considered as statistically significant.

## Search results

The systematic literature search was terminated on October 31st, 2019, and next the titles of the articles, keywords as well as abstracts were assessed for eligibility. In the first screening, two independent researchers, blinded to each other, reviewed retrieved 894 studies, and subsequently removed 471 duplicates. 10 questionable papers were discussed and the decision was made together after negotiation (kappa statistic, 0.96). Since 273 papers did not meet the PRISMA criteria, 150 eligible papers were finally selected.

Compliance with the inclusion and exclusion criteria for the meta-analysis reduced the number of eligible papers to 43. Since 26 papers out of originally chosen 43 did not include appropriate data for the analysis of the effect of selected biological agents and a small molecule on HRQOL (e.g. changes in the DLQI scoring prior to and after the commencement of therapy, comparison between placebo and drug effect), while others lacked biologic agents in their analysis (phototherapy, topical therapy, methotrexate or other non-biologic drugs were explored), we were left with only 17 eligible papers.

However, we decided to include all of the 43 papers in our analysis in order to demonstrate the differences in the DLQI scoring for different races.

The detailed strategy used for the literature identification, screening and selection is summarized in Fig 1.

## Results

### HRQOL measured by the DLQI–study characteristics

In the 43 studies which met our inclusion criteria the total of 25,898 individuals were evaluated by the DLQI and the weighted mean scores were derived for the analysis. The performed meta-analysis took into account the patients' age, sex and race in order evaluate the mean differences in the DLQI scores. Among the enrolled patients, we observed the predominance of

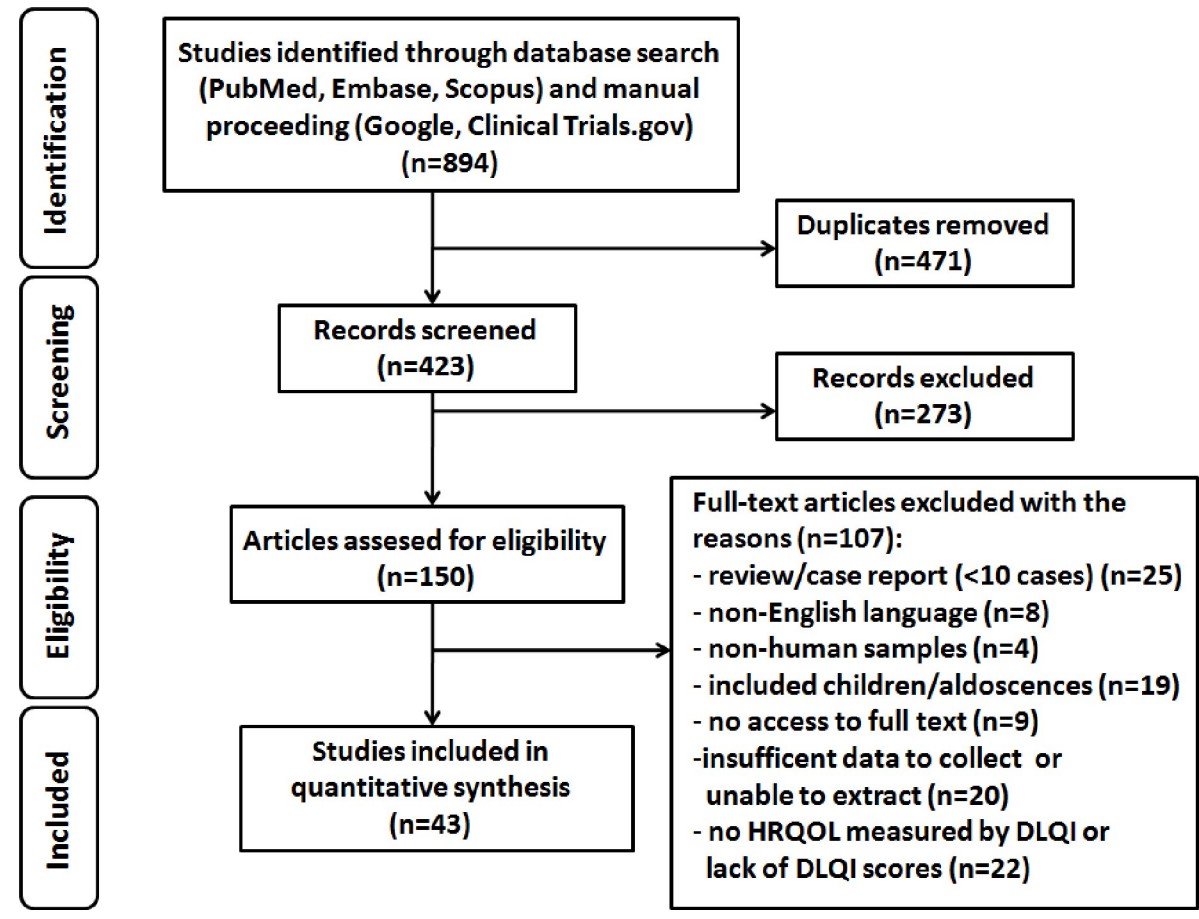

**Fig 1. Flowchart of the research methodology for systematic review based on the PRISMA guidelines.**

male plaque psoriatic patients (range from 50 to 81%) and the individuals were on average over 45 years old. However, some of the studies enrolled individuals of different ages (i.e. 18–82 years old), and even focused on elderly psoriatic subjects (>65 years old) [15–18]. Some of the papers revealed almost equal proportions between male and female study subjects although at times the percentage of females was higher than the percentage of males [19–23]. As for the race of the evaluated patients, most of the studies demonstrated differentiation in the patients' race. Only 13 out of the analysed 43 studies evaluated single-race groups (8 studies enrolled only Asians and 5 studies enrolled only Caucasians). Moreover, another 11 out of the analysed 43 papers suffered from the lack of data concerning the patients' race. The mean DLQI scores with 95%CI achieved in selected papers are presented in Table 1.

The mean DLQI scores ranged from 6.83 to 17.8 throughout the studies. The overall DLQI score in the model was 12.12 (95%CI: 11.24 to 13.06). A random-effects model demonstrated significant considerable heterogeneity of the study results ($I^2$ = 98%; p<0.001) (S1A Fig). In order to examine bias in the results of the performed meta-analysis, the funnel plot analysis and LFK index were applied. Despite considerable heterogeneity of the results, we did not find asymmetry in the funnel plot graph (S1B Fig), which confirmed the LFK score of 0.86 (S1C Fig).

## Changes in the DLQI score as an effect of selected agents (treatment vs baseline)

Out of the selected studies 17 papers comprising 7,466 subjects were eligible for evaluation of the effect of selected agents on DLQI scoring. The values of SMD were compared between the groups of patients screened by the DLQI prior to the therapy and after the commencement of treatment. The detailed data referring to the effect of selected agents on the DLQI scoring and SMD (Hedge's g) are presented in Table 2. Due to a small number of studies (<10) for each therapy regimen, the asymmetry was measured by the LFK index and a random-effects model was applied.

**Infliximab.**   Out of the selected agents, infliximab was the most frequently studied drug, which was reported in 7 papers with the total number of 2,416 patients. These 7 studies demonstrated considerable heterogeneity ($I^2$ = 92%; p = <0.01) and minor asymmetry (LFK = 1.47). All the studies demonstrated a large effect size of infliximab that ranged from -0.98 to -1.82, and the pooled g was -1.49 (95%CI: -1.74 to -1.25). The duration of therapy varied from 10 to 42 weeks (S2C Fig).

**Adalimumab.**   The effect of adalimumab on the improvement of the DLQI scoring was evaluated in 4 studies (1,632 patients) (S2A Fig). All of them demonstrated a large effect size on the DLQI with the pooled overall Hedge's g (g) of -1.34 (95%CI: -1.52 to -1.15) and considerable heterogeneity ($I^2$ = 76%; p = 0.01). The selected studies enrolled over 80% participants of the Caucasian race. The therapy outcomes were collected after either 12 or 16 or 24 weeks of the treatment. The LFK index of 2.0 demonstrated minor asymmetry of the results.

**Ixekizumab.**   The effect of ixekizumab on the HRQOL improvement measured by the DLQI was analyzed in 3 studies (1,148 patients), however, one of them evaluated two therapy regimens depending on the frequency of the drug administration (once every two weeks or once every four weeks) (Imafuku et al.) [24] (S2B Fig). Two studies enrolled only Asian subjects and, on the whole, the therapy duration ranged from 12 to 24 weeks. The overall pooled effect of ixekizumab reached g of -1.39 (95%CI: -1.61 to -1.17) with considerable heterogeneity ($I^2$ = 80%; p = <0.01) and major asymmetry (LFK = 2.82).

**Secukinuma.**   The data concerning secukinumab efficacy was pooled from two papers summarizing the phase III trials: ERASURE, FIXTURE, CLEAR and PRIME, including 946 psoriatic individuals [25]. Considerable heterogeneity was found between the papers ($I^2$ =

**Table 1. Summary results of the meta-analysis on health-related quality of life (HRQOL) in plaque psoriasis adult patients measured by the dermatology life quality index (DLQI) depending on the race.**

| Study/trial name | Race | Mean DLQI | Lower limit | Upper limit | Sample size |
|---|---|---|---|---|---|
| **Armstrong (2019)** VOYAGE 1&2 | Caucasian (81.9%) | 14.5 | 11.9 | 15.4 | 1829 |
| | Asian (14.1%) | | | | |
| | Afroamerican (4.0%) | | | | |
| **Arora (2018)** | Asian (100%) | 10.68 | 8.67 | 12.7 | 29 |
| **Körber (2018)** ERASURE, FIXTURE, CLEAR | Caucasian (77.1%) | 13.8 | 13.3 | 14.3 | 841 |
| | Asian (15.7%) | | | | |
| | Afroamerican and unknown (7.2%) | | | | |
| **Papadavid (2018)** | Caucasian (100%) | 11.1 | 8.84 | 13.4 | 50 |
| **Petridis (2018)** | Caucasian (100%) | 15.0 | 13.6 | 15.4 | 136 |
| **Zachariae (2018)** | Unknown | 10.9 | 9.88 | 11.9 | 142 |
| **Imafuku (2017)** | Caucasian >90% | 13.0 | 12.4 | 13.6 | 1296 |
| **Guenther (2017)** UNCOVER 2&3 | Caucasian (92.4%) | 12.5 | 12.2 | 12.8 | 2570 |
| | Asian (3.1%) | | | | |
| | Afroamerican and other (4.5%) | | | | |
| **Parthasaradhi (2017)** | Asian (100%) | 15.43 | 14.7 | 16.2 | 155 |
| **Chua (2017)** | Asian (100%) | 10.3 | 7.18 | 13.04 | 28 |
| **Strober (2017)** | Unknown | 11.0 | 10.1 | 11.9 | 221 |
| **Breteque (2017)** | Unknown | 11.3 | 10.8 | 11.8 | 749 |
| **Thaçi (2017)a** ESTEEM1 | Caucasian >90% | 12.4 | 11.9 | 12.9 | 844 |
| **Thaçi (2017)b** ESTEEM2 | Caucasian >90% | 12.6 | 11.9 | 13.3 | 411 |
| **Sticherling (2017)** PRIME | Unknown | 17.8 | 16.8 | 18.8 | 105 |
| **Blauvelt (2017)** CLEAR | Caucasian (86.8%) | 13.3 | 12.7 | 13.9 | 676 |
| | Other (13.2%) | | | | |
| **Griffiths (2017)** | Caucasian (92.2%) | 12.6 | 12.0 | 12.8 | 666 |
| | Other (7.8%) | | | | |
| **Kimbal (2016)a** JAPD | Caucasian (79.6%) | 9.3 | 8.39 | 10.2 | 270 |
| | Asian (16.7%) | | | | |
| | Afroamerican and other (3.7%) | | | | |
| **Kimbal (2016)b** RHAZ | Caucasian (92.5%) | 13.1 | 12.7 | 13.5 | 1294 |
| | Asian (4.8%) | | | | |
| | Afroamerican and other (2.7%) | | | | |
| **Kimbal (2016)c** RHBC | Caucasian (92.7%) | 12.0 | 11.6 | 12.4 | 1340 |
| | Asian (3.0%) | | | | |
| | Afroamerican and other (4.3%) | | | | |
| **Kimbal (2016)d** RHBA | Caucasian (92.6%) | 12.3 | 11.9 | 12.7 | 1218 |
| | Asian (3.0%) | | | | |
| | Afroamerican and other (4.4%) | | | | |
| **Torii (2016)** | Asian (100%) | 9.29 | 8.53 | 10.0 | 314 |
| **Maccari (2016)** LIBERE | Unknown | 10.0 | 9.05 | 10.9 | 125 |
| **Valenzuela (2016)** | Unknown | 12.8 | 12.3 | 13.3 | 1101 |
| **Atakan (2016)** TUR-PSO | Caucasian (100%) | 6.83 | 6.62 | 7.04 | 3131 |
| **Ayala (2015)** TANGO | Caucasian (97.4%) | 12.9 | 10.6 | 15.3 | 38 |
| | Other (2.6%) | | | | |
| **Gahalaut (2014)** | Asian (100%) | 14.45 | 13.5 | 15.4 | 40 |
| **Zhu (2014)** | Unknown | 10.9 | 9.88 | 11.9 | 142 |

*(Continued)*

**Table 1.** (Continued)

| Study/trial name | Race | Mean DLQI | Lower limit | Upper limit | Sample size |
|---|---|---|---|---|---|
| **Poulin (2014)** | Caucasian (92.5%) | 11.6 | 10.7 | 12.5 | 240 |
| | Other (7.5%) | | | | |
| **Larsen (2013)** | Unknown | 11.4 | 10.3 | 12.5 | 163 |
| **Roongpisuthipong (2013)** | Asian (100%) | 8.0 | 5.52 | 10.5 | 10 |
| **Vender (2012)** | Unknown | 13.7 | 12.9 | 14.5 | 246 |
| **Nakagawa (2012)** | Asian (100%) | 10.9 | 9.9 | 11.9 | 158 |
| **Yang (2012)** | Asian (100%) | 14.4 | 12.9 | 16.0 | 129 |
| **Barker (2011)** RESTORE | Caucasian (97.5%) | 13.6 | 12.9 | 14.4 | 868 |
| | Other (2.5%) | | | | |
| **Torii (2010)** | Asian (100%) | 11.6 | 8.9 | 14.2 | 54 |
| Revicki (2007) | White (90.8%) | 11.3 | 10.9 | 11.7 | 1205 |
| | Asian (2.2%) | | | | |
| | Other (7.0%) | | | | |
| **Flystrom (2008)** | Unknown | 8.6 | 6.6 | 10.6 | 68 |
| **Colombo (2008)** | Caucasian (100%) | 8.9 | 7.94 | 9.86 | 150 |
| **Feldman (2008)** | Caucasian (92.6%) | 13.1 | 12.6 | 13.6 | 835 |
| | Asian (2.8%) | | | | |
| | Afroamerican and Other (4.6%) | | | | |
| **Shikiar (2007)** | Caucasian (90%) | 13.5 | 12.0 | 15.0 | 95 |
| | Other (10%) | | | | |
| **Shikiar (2006)** | Caucasian (90.5%) | 12.71 | 11.6 | 13.9 | 147 |
| | Asian (3.4%) | | | | |
| | Afroamerican (2.7%) | | | | |
| | Other (3.4%) | | | | |
| **Reich (2006)** | Unknown | 12.5 | 11.8 | 13.2 | 378 |
| **Feldman (2005)** | Unknown | 12.7 | 11.7 | 13.7 | 198 |
| **Feldman (2004)** | Unknown | 11.58 | 10.5 | 12.5 | 182 |
| **Gordon (2003)** | Caucasian (90%) | 12.0 | 11.4 | 12.6 | 556 |
| | Other (10%) | | | | |
| **Finlay (2003)** | Caucasian (90%) | 11.1 | 10.5 | 12.1 | 474 |
| | Other (10%) | | | | |

95%; p<0.001), and the pooled size effect for secukinumab was -2.35 (95%CI: -3.25 to -1.44). Because only two papers were pooled, the asymmetry was not possible to assess. Both papers differed in the duration of treatment, and they were as follows: 24 and 52 weeks, respectively [23, 26] (S2D Fig).

**Tofacitinib.** Two phase III trials (1,324 patients) reported mean DLQI changes resulting from tofacitinib therapy. Griffiths et al. studied two therapy regimens dependent on a drug dose (5 or 10 mg of tofacitinib), and Valenzuela et al. used the same protocol [27, 28]. However, therapy duration was 12 and 24 weeks, respectively. In the analysis the overall pooled g was -1.27 (95%CI: -1.52 to -1.02) (S2D Fig). The $I^2$ score was equal to 89% (p<0.01), while the LFK = -2.32 (major asymmetry).

## Changes in the DLQI score as an effect of selected agents (treatment vs placebo)

In 10 clinical trials (3 –phase II and 7 –phase III trials) the selected agents were used as a comparator to a placebo in order to assess the changes in HRQOL measured by the DLQI (Table 3).

**Table 2. Summary of the published data and standardized mean differences (SMDs; Hedge's g) of DLQI measured prior to and after the commencement of infliximab, adalimumab, ixekizumab, secukinumab and tofacitinib plaque psoriasis patients.**

| Study | Drug | Patients (n) | Treatment duration (weeks) | DLQI prior to treatment* | DLQI after the treatment* | SMD (Hedge's g) | Citation |
|---|---|---|---|---|---|---|---|
| Torii (2016) | Infliximab | 314 | 42 | 9.29±6.85 | 3.61±4.18 | -0.98 | [33] |
| Yang (2012) | | 80 | 26 | 14.4±6.2 | 4.3±5.6 | -1.71 | [29] |
| Barker (2011) | | 653 | 26 | 13.5±7.2 | 2.2±5 | -1.82 | [32] |
| Torii (2010) | | 35 | 42 | 12.7±6.8 | 2.7±8 | -1.35 | [16] |
| Feldman (2008) | | 835 | 10 | 13±7 | 3.1±5.1 | -1.62 | [54] |
| Reich (2006) | | 301 | 24 | 12.7±7 | 2.7±7.3 | -1.40 | [38] |
| Feldman (2005) | | 198 | 10 | 12.7±7 | 3.1±5.1 | -1.57 | [39] |
| Pooled SMD (Hedge's g) of -1.49 (95%CI: -1.74 to -1.25); I² = 92% | | | | | | | |
| Pooled effect estimates of -1.53 (95%CI: -1.81 to -1.24); I² = 92% | | | | | | | |
| Armstrong (2019) | Adalimumab | 582 | 24 | 14.6±7.1 | 5.0±7.7 | -1.27 | [36] |
| Revicki (2007) | | 808 | 16 | 11.3±6.6 | 3.0±4.5 | -1.47 | [15] |
| Shikiar (2007) | | 95 | 12 | 13.5±7.7 | 2.4±6.3 | -1.58 | [30] |
| Shikiar (2006) | | 147 | 12 | 12.71±7.18 | 5.28±6.49 | -1.09 | [34] |
| Pooled SMD (Hedge's g) of -1.34 (95%CI: -1.52 to -1.15); I² = 76% | | | | | | | |
| Pooled effect estimates of -1.45 (95%CI: -1.68 to -1.22); I² = 80% | | | | | | | |
| Zachariae (2018) | Ixekizumab | 142 | 24 | 10.9±6.2 | 2.5±5.8 | -1.40 | [37] |
| Imafuku (2017)a | | 432 | 12 | 13.2±7 | 2.5±6.91 | -1.54 | [24] |
| Imafuku (2017)a | | 433 | 12 | 13.9±7 | 2.8±7.38 | -1.54 | [24] |
| Zhu (2014) | | 141 | 16 | 10.9±6.2 | 4.7±6.1 | -1.01 | [35] |
| Pooled SMD (Hedge's g) of -1.39 (95%CI: -1.61 to -1.17); I² = 80% | | | | | | | |
| Pooled effect estimates of -1.36 (95%CI: -1.56 to -1.16); I² = 80% | | | | | | | |
| Körber (2018) | Secukinumab | 841 | 52 | 13.8±7.4 | 2.2±4.4 | -1.91 | [23] |
| Sticherling (2017) | | 105 | 24 | 17.8±5.3 | 2.8±5.3 | -2.83 | [26] |
| Pooled SMD (Hedge's g) of -2.35 (95%CI: -3.25 to -1.44); I² = 95% | | | | | | | |
| Pooled effect estimates of -1.98 (95%CI: -3.12 to -1.84); I² = 95% | | | | | | | |
| Griffiths (2017)a | Tofacitinib | 331 | 24 | 12.4±6.2 | 5.1±7.28 | -1.08 | [27] |
| Griffiths (2017)b | | 335 | 24 | 12.4±6.2 | 2.6±7.28 | -1.45 | [27] |
| Valenzuela (2016) a | | 328 | 12 | 13±7.2 | 5.6±7.24 | -1.02 | [28] |
| Valenzuela (2016) b | | 330 | 12 | 13.3±7.27 | 3.5±5.45 | -1.53 | [28] |
| Pooled SMD (Hedge's g) of -1.27 (95%CI: -1.61 to -1.17); I² = 89% | | | | | | | |
| Pooled effect estimates of -1.21 (95%CI: -1.49 to -1.23); I² = 89% | | | | | | | |

* mean±SD.

Among them, the model for infliximab enrolled 50% of all the eligible studies (S3A Fig). The overall pooled effect of infliximab compared to placebo was -0.91 (95%CI: -1.41 to -0.91) ($I^2$ =, p = <0.01), but Yang et al. and Torii et al. studies showed a small effect size, as follows: -0.21 and -0.01, respectively [16, 29]. Both of the studies involved only 183 patients in total, which probably affected the major asymmetry of the results represented by the LFK index (4.73). The studies concerning the comparison of therapy efficacy (improvement of the DLQI scoring) between adalimumab and placebo demonstrated the lack of heterogeneity ($I^2$ = 5%, p = 0.35) (S3B Fig). The pooled effect size was -1.11 (95%CI: -1.20 to -1.02). The study of Shikiar et al. [30] weighted only 6.3% due to a small number of enrolled subjects (95 patients). Both ixekizumab and tofacitinib had a large effect size on the DLQI: -1.19 and -0.86,

**Table 3. Summary of the published data and standardized mean differences (SMDs; Hedge's g) of DLQI between two groups of plaque psoriasis individuals treated with the use of either a selected agent or a placebo.**

| Study/phase | Drug | Drug treated group (n) | Placebo treated group (n) | Reduction in DLQI from baseline to outcome (drug)* | Reduction in DLQI from baseline to outcome (placebo)* | SMD (Hedge's g) | Citation |
|---|---|---|---|---|---|---|---|
| Yang (2012) III | Infliximab vs placebo | 80 | 42 | -10.1 | -8.9 | -0.21 | 29 |
| Torii (2010) III | | 35 | 19 | -10.0 | -7.7 | -0.01 | 21 |
| Feldman (2008) II | | 835 | 208 | -9.9 | -0.6 | -1.45 | 54 |
| Reich (2006) III | | 301 | 77 | -10.0 | -0.2 | -1.26 | 38 |
| Feldman (2005) II | | 198 | 51 | -9.6 | -2.6 | -1.44 | 39 |
| Pooled SMD (Hedge's g) of -0.91 (95%CI: -1.41 to -0.42); I² = 93% | | | | | | | |
| Pooled effect estimates of -1.12 (95%CI: -1.82 to -0.86); I² = 93% | | | | | | | |
| Armstrong (2019) III | Adalimumab vs placebo | 582 | 422 | -9.5 | -1.8 | -1.06 | 36 |
| Revicki (2007) III | | 808 | 397 | -8.3 | -2.2 | -1.13 | 20 |
| Shikiar (2007) II | | 95 | 52 | -11.1 | -1.5 | -1.33 | 30 |
| Pooled SMD (Hedge's g) of -1.11 (95%CI: -1.20 to -1.02); I² = 5% | | | | | | | |
| Pooled effect estimates of -1.11 (95%CI: -1.20 to -1.02); I² = 5% | | | | | | | |
| Imafuku (2017) a III | Ixekizumab vs placebo | 432 | 431 | -10.7 | -2.6 | -1.22 | 24 |
| Imafuku (2017) b III | | 433 | 431 | -11.1 | -2.6 | -1.15 | 24 |
| Pooled SMD (Hedge's g) of -1.19 (95%CI: -1.29 to -1.09); I² = n/a | | | | | | | |
| Pooled effect estimates of -1.19 (95%CI: -1.29 to -1.09); I² = n/a | | | | | | | |
| Valenzuela (2016)a III | Tofacitinib vs placebo | 328 | 107 | -7.4 | -2.0 | -0.63 | 28 |
| Valenzuela (2016)b III | | 330 | 107 | -9.8 | | -1.09 | 28 |
| Pooled SMD (Hedge's g) of -0.86 (95%CI: -1.31 to -0.40); I² = n/a | | | | | | | |
| Pooled effect estimates of -0.86 (95%CI: -1.31 to -0.40); I² = n/a | | | | | | | |

* mean change.

respectively, compared to placebo. The data were derived from single studies with the analysis of subgroups depending on the used dose (Table 3).

## Discussion

The presented meta-analysis was based on the DLQI weighted mean scores derived from 43 studies meeting the inclusion criteria and comprising 25,898 adults suffering from plaque psoriasis of the Caucasian, Asian, Afro-American or unknown race investigated in single-race or mixed-race groups. As for the studied race, apart from Atakan's et al. research into a group of Turks with plaque psoriasis, the obtained results were similar, i.e. no asymmetry was observed (the LFK score of 0.86) [31]. Atakan et al. study results may have been different because of different geographic and climatic regions the investigated Turkish patients lived in. As for the other investigated race groups, it is difficult to prove any racial influence on the DLQI because of the lack of information regarding their geographic and climatic conditions. In general, the race of the studied subjects does not give grounds for making an assumption that the race had

an impact on the improvement of the health-related quality of life (HRQOL) of adult plaque psoriatic patients treated with the investigated agents.

Out of the aforementioned 43 studies regarding adult plaque psoriatic patients, 17 papers comprising 7,466 study subjects were further analyzed with reference to the effects of selected biologics, i.e. infliximab, adalimumab, ixekizumab, and secukinumab, and a small molecule, i.e. tofacitinib, on the HRQOL of the studied patients. The $I^2$ test results for all the studied agents revealed high heterogeneity, therefore a random-effects model was used. Out of all the investigated agents, secukinumab had the highest heterogeneity ($I^2 = 95\%$), whereas the lowest heterogeneity was observed for adalimumab ($I^2 = 76\%$). Apparently, the use of these agents positively affected the studied patients' quality of life, which was confirmed by the Pooled standardized mean differences (SMD; Hedge's g score) values. Interestingly, the higher the DLQI score prior to the commencement of the treatment with infliximab, adalimumab, ixekizumab, secukinumab and tofacitinib, the better the improvement of the patients' quality of life, which was confirmed by the SMD; (Hedge's g).

The majority of scientific studies (7 papers) included in this meta-analysis investigated the effect of infliximab on the HRQOL of adult plaque psoriatic patients (2,416 individuals). While the best improvement of the quality of life (QqL) after commencement of infliximab therapy in the group consisting of 97.5% of Caucasians was reported by Barker et al (the SMD; Hedge's g: -1,82), the least improvement in the quality of life was observed in both Torri et al. studies, where all the treated patients were Asians [16, 32, 33].

According to the DLQI, the best effect of treatment with adalimumab was observed in the study of Shikiar et al. [30] of 2007, where the value of the DLQI before the treatment was 13.5 ±7.7 and after the treatment it dropped to 2.4±6.3, the number of studied patients, however, was smaller than in the other studied groups treated with this agent (the SMD; Hedge's g of -1.58) [30]. Yet, the Pooled SMD (Hedge's g) for adalimumab was -1.34, which was indicative of its lower efficacy in the improvement of the HRQOL of the studied adult plaque psoriatic patients in comparison with the other biologic agents. The duration of treatment (in weeks) with adalimumab had only a slight effect on the improvement of the HRQOL of the studied patients, however, its dosage and frequency of administration proved to play a role. In comparison to the Shikiar et al. study results of 2006 (the SMD; Hedge's g: - 1.09), in which the patients were administered 80 mg of adalimumab at week 0 and week 1, followed by 40 mg every week beginning at week 2, the Shikiar et al. study results of 2007, where the patients received 80 mg of adalimumab at 0 week, followed by 40 mg every other week beginning at week 1, showed better improvement of the HRQOL of the investigated patients (the SMD; Hedge's g: - 1.58) [30, 34].

The values of the SMD (Hedge's g) for the effects of treatment with ixekizumab did not differ much from each other except for the SMD value obtained in the Zhu et al. study (the SMD; Hedge's g: -1.01), which meant that the improvement of the HRQOL in the studied patients was worse in comparison to the other analyzed studies [35]. This may be connected with the administered doses of ixekizumab in two groups investigated in the Zhu et al. study, i.e.10 mg/ 25mg in one group and 75/150mg in the other group [35]. The dosage in the former group may have had a negative effect on the final values of the SMD (Hedge's g). The study of Imafuku et al. of 2017 reported an improved HRQOL after a 12-week therapy with ixekizumab (the SMD; Hedge's g -1.54), which may have been connected with the dosage and frequency of ixekizumab administration [24]. The dosage of ixekizumab was higher in Imafuku et al. than in Zhu et al study [24, 35].

The best value of the Pooled SMD (Hedge's g), i.e. -2.35, was obtained by secukinumab, which means that the treatment with this agent was most effective and improved the HRQOL of the patients the most. However, this good result might have been caused by the fact that

only two groups of patients were investigated, i.e. the Korber et al. group—841 patients, and the Sticherling et al. group—105 patients, where the latter had the highest DLQI value (17.8 ±5.3) prior to the commencement of the treatment, whereas after the treatment the value of the DLQI dropped to 2.8±5.3 (the SMD; Hedge's g– 2.83) [23, 26].

Quite recently, Jabbar-Lopez et al. performed a network meta-analysis comparing biologics (adalimumab, etanercept, infliximab, secukinumab, ustekinumab, and ixekizumab) with one another, methotrexate, or placebo in the patients with moderate-severe chronic plaque psoriasis taking into account not only the mean change in the DLQI but also assessing other outcomes, i.e. clear/nearly clear and PASI 75 [36]. All biologic therapies had statistically significant increased odds of the mean change in the DLQI compared with placebo at 12 to 16 weeks. Secukinumab performed the best and placebo the worst in terms of the mean change in the DLQI. Adalimumab, infliximab, ixekizumab, secukinumab, and ustekinumab were all similar with regard to the mean change in the DLQI, whereas etanercept was less effective [36].

Griffiths et al. and Valenzuela et al. did research into the efficacy of tofacitinib, a synthetic small molecule, with reference to its ability to improve the HRQOL of adult plaque psoriatic patients [27, 28]. Even though Valenzuela et al. reported a relatively good effect of tofacitinib on the improvement of the psoriatic patients' quality of life, its overall efficacy is the poorest of all the investigated agents (the Pooled SMD; Hedge's g of -1.27) [28]. Tian et al. performed a meta-analysis to evaluate the efficacy and safety of tofacitinib in the patients with chronic plaque psoriasis, however, they assessed outcomes other than the DLQI, i.e. physician global assessment, PASI75 and PASI90 [37]. The authors found that compared to placebo tofacitinib was significantly more effective, whereas the incidence of adverse reactions was significantly higher [37].

In the presented meta-analysis, a comparison between the effect of treatment with infliximab, adalimumab, ixekizumab, and tofacitinib versus placebo on the improvement of the HRQOL of adult plaque psoriatic patients showed varied heterogeneity. The studies comparing adalimumab vs placebo demonstrated the lack of heterogeneity ($I^2 = 5\%$), while the analysis of infliximab vs placebo presented high heterogeneity ($I^2 = 93\%$). The lack of heterogeneity in the analysis of adalimumab vs placebo may have been related to a small number of the investigated groups, their predominant Caucasian race, and similar SMD; (Hedge's g) [15, 30, 38]. The highest heterogeneity observed in the analysis comparing the effect of infliximab vs placebo may have resulted from dissimilar standardized mean differences = SMD (from Hedge's g = -0,21 to Hedge's g = -1,45), great differences in the number of patients taking the drug or placebo and various races in each studied group. The $I^2$ value of ixekizumab and tofacitinib versus placebo was not calculated because of the insufficient study material [24, 28].

Generally, the effect of each of the applied agents on the HRQOL of the studied patients was considerably better in comparison with the use of placebo, i.e. the Pooled SMD; Hedge's g of -1.11 for adalimumab, -1.19 for ixekizumab, -0.86 for tofacitinib. Interestingly, the Pooled SMD; Hedge's g was -0.91 for infliximab and the placebo effect was remarkably high in the study of Yang et al. (the SMD; Hedge's g -0.21) and in the study of Torii et al. (the SMD, Hedge's g -0.01) [16, 29]. It is worth noting that in the Yang et al. and Torii et al. studies the investigated groups were 100% Asian, which could be suggestive of some racial significance [16, 29]. However, since both Yang's and Torii's groups were small (n = 80, n = 35, respectively) the validity of these results should be confirmed by further research [16, 29].

## Strengths and limitations of the study

Limitations of this meta-analysis result from the use of solely DLQI to assess the effect of selected biologics and a small molecule on Health-Related Quality of Life (HRQoL) in adult

plaque psoriasis patients. Heterogeneity was noted across the studies. Apart from the lack of information about the race of the studied patients in certain analyzed papers, the meta-analysis includes limited information on the patients' sex, age, duration of psoriasis and its onset as well as co-existing diseases [24, 26, 28, 35, 39–41]. Thus, the data quality and heterogeneity may restrict the interpretation of the pooled risk estimates.

## Conclusion

Infliximab, adalimumab, ixekizumab, secukinumab and tofacitinib in adult plaque psoriatic patients improved HRQOL measured by the DLQI. The patients with lower quality of life before treatment obtained better results. To the best of our knowledge, it is the first meta-analysis that compares the effects of the selected biologics and a small molecule on HRQOL measured by the DLQI in adult plaque psoriasis patients. This meta-analysis also compares biologics, a small molecule and the placebo effect on the DLQI in psoriatic patients.

## Supporting information

**S1 File. PRISMA 2009 checklist.**
(DOC)

**S2 File. PRISMA NMA checklist of items to include when reporting a systematic review involving a network meta-analysis.**
(DOCX)

**S1 Fig.**
(TIF)

**S2 Fig.**
(TIF)

**S3 Fig.**
(TIF)

## Author Contributions

**Conceptualization:** Anna Karpińska-Mirecka, Joanna Bartosińska, Dorota Krasowska.

**Data curation:** Anna Karpińska-Mirecka, Joanna Bartosińska, Dorota Krasowska.

**Formal analysis:** Anna Karpińska-Mirecka, Joanna Bartosińska, Dorota Krasowska.

**Investigation:** Anna Karpińska-Mirecka, Dorota Krasowska.

**Methodology:** Anna Karpińska-Mirecka, Joanna Bartosińska.

**Project administration:** Joanna Bartosińska, Dorota Krasowska.

**Resources:** Anna Karpińska-Mirecka.

**Supervision:** Joanna Bartosińska, Dorota Krasowska.

**Validation:** Anna Karpińska-Mirecka, Joanna Bartosińska.

**Visualization:** Anna Karpińska-Mirecka, Dorota Krasowska.

**Writing – original draft:** Anna Karpińska-Mirecka.

**Writing – review & editing:** Anna Karpińska-Mirecka, Joanna Bartosińska, Dorota Krasowska.

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
