## [Decision Letter · Decision Letter 0]

10 Aug 2020

PONE-D-20-16022

The effects of selected biologics and a small molecule on Health-Related Quality of Life (HRQoL) in adult plaque psoriasis patients measured by Dermatology Life Quality Index (DLQI) – systematic review and meta-analysis

PLOS ONE

Dear Dr. Karpinska-Mirecka,

Thank you for submitting your manuscript to PLOS ONE. After careful consideration, we feel that it has merit but does not fully meet PLOS ONE’s publication criteria as it currently stands. Therefore, we invite you to submit a revised version of the manuscript that addresses the points raised during the review process.

We look forward to receiving your revised manuscript.

Kind regards,

Ahmed Negida, MD

Academic Editor

PLOS ONE

Journal Requirements:

2. Please ensure you have included the full electronic search strategy for at least one database and uploaded it as an additional file.

Reviewers' comments:

Reviewer's Responses to Questions

**Comments to the Author**

1. Is the manuscript technically sound, and do the data support the conclusions?

Reviewer #1: Yes

Reviewer #2: Yes

Reviewer #3: Yes

Reviewer #4: Partly

Reviewer #5: Yes

2. Has the statistical analysis been performed appropriately and rigorously? 

Reviewer #1: Yes

Reviewer #2: Yes

Reviewer #3: Yes

Reviewer #4: Yes

Reviewer #5: Yes

3. Have the authors made all data underlying the findings in their manuscript fully available?

Reviewer #1: Yes

Reviewer #2: Yes

Reviewer #3: Yes

Reviewer #4: Yes

Reviewer #5: Yes

4. Is the manuscript presented in an intelligible fashion and written in standard English?

Reviewer #1: Yes

Reviewer #2: Yes

Reviewer #3: No

Reviewer #4: No

Reviewer #5: Yes

5. Review Comments to the Author

Reviewer #1: The present meta-analysis aims to assess the utility of biological agents and one small molecule inhibitor (Tocafitinib) in improving the quality of life of Psoriasis patients (as measured by the DLQI). Psoriasis can be a highly debilitating condition, and the advent of a number of new therapeutic options does call into need a systematic analysis of the different treatments. There is a recent network meta-analysis (2017) by Lopez et al. covering the same biological agents (infliximab, adalimumab, ixekizumab, Secukinumab). Due to the network structure of the analysis, it was able to better compare the different biological agents. In addition, it had a somewhat wider coverage of outcomes, as it was not restricted solely to the DLQI as an outcome measure. There is also one recent meta-analysis (2019) by Tian et al. conducted specifically on the small molecule inhibitor Tocafitinib (assessing outcomes other than the DLQI; namely Physician global assessment response, PASI75 and PASI90) in which a significant improvement was seen.

In addition, I believe correcting the following points of methodology would probably improve the overall value of the manuscript and hopefully improve the chances of a successful publication:

1. The authors state that 423 studies qualified for full-text screening; however, the PRISMA diagram shows that only 150 articles were assessed for eligibility.

2. The authors’ PRISMA diagram shows a total of 43 included studies; however, they then proceed to choose 17 papers for evaluation of select biologic agents without clarifying the rationale as to why these 17 studies were the chosen ones. This may well be justified due to the nature of the selected studies, but I believe it would be best for the authors to clearly outline the justification for their selection.

3. The authors included a network diagram despite the fact that no network analysis was done. I think, for the sake of brevity, the diagram should probably be removed.

4. I do not believe the “citations” column in tables 1-3 serve to add much value, as the number of citations of included studies is not directly relevant to the review.

In addition, as the quality of writing greatly affects publication chances, and as the manuscript contains some linguistic errors, the paper would probably benefit from a grammatical revision to improve readability.

The aforementioned previous Meta-Analyses:

1. Quantitative Evaluation of Biologic Therapy Options for Psoriasis: A Systematic Review and Network Meta-Analysis

2. Efficacy and Safety of Tofacitinib for the Treatment of Chronic Plaque Psoriasis: A Systematic Review and Meta-Analysis

Reviewer #2: This is a systematic review and meta-analysis in which the authors investigates the effects and efficiency of different biologics and many biological drugs on Health-Related Quality of Life (HRQoL) in adult plaque psoriasis patients measured by Dermatology Life Quality Index (DLQI). DLQI is commonly used tool to assess quality of life of patients with skin diseases. The authors found that Infliximab, adalimumab, ixekizumab, secukinumab and tofacitinib in adult plaque psoriatic patients improved HRQOL measured by DLQI. The manuscript is well-written and structured. It also provides a tool for the physician to choose the right biological agent to improve the quality of life in patients with psoriasis.

Reviewer #3: The authors performed a systematic review and meta-analysis to investigate the effect of infliximab, adalimumab, ixekizumab, secukinumab, and tofacitinib on Health-Related Quality of Life (HRQOL) measured by DLQI in adult plaque psoriatic patients.

Although the overall approach of this review is proper, I have few comments:

1. Title: abbreviations should be avoided. Also, replace (– systematic) with (: A systematic).

2. The short title is missing.

3. Abstract

a) Main Objectives: remove (Aim of the study)

b) Material and Methods

- Searched through XX, insert the last date searched.

- clearly outline the eligibility criteria

- Insert the software used for analysis.

c) The results are inadequately reported, please add data of the pooled effect estimates.

d) Keywords are missing.

4. Introduction: place references at the end of sentences.

5. Methods

- The authors searched databases until October 2019. I would suggest updating the searching process for recently published relevant articles.

- No data on how the risk of bias assessment was performed.

6. Results

- Data on the screening process are incomplete and wrongly placed. Please move it to “Methods.”

- Data in Table 1 and Figure 2 are the same. Repeating data is not preferred; thus, I would recommend that one of them can be submitted as a supplementary file. The same is true for other tables and figures of the same outcome.

7. A completed PRISMA checklist is missing.

8. Figures should be submitted in better resolutions.

9. Language: The entire manuscript needs extensive professional revision for grammatical errors and stylistic editing to improve the quality of English. For example,

- Page 1: (to assess the quality), not (to assess quality); (confirm the positive effect) not (confirm positive effect); (with an overall DLQI score) not (with overall DLQI score); (The random-effects model) not (Random effects model). (Scopus, ClinicalTrials.gov and manual searching) & (ixekizumab, secukinumab and tofacitinib), a comma should be placed before (and), etc.

Reviewer #4: Dear Author,

There are some points to consider when you revise your manuscript.

The title of your study is too long. It would be better to make it short such as following;

(The effects of selected biologics and a small molecule on Health-Related Quality of Life: systematic review and meta-analysis.)

The affiliation sentence needs revision.

The reference need to be revised according to the journal guideline.

It also need English editing.

Why did you include 43 studies while only 17 of them assessed the DQLI of patients on biologic treatments? based on your objective you should only include these 17 studies.

Reviewer #5: Title

I believe it is unnecessary to put in the title of the article that Health-Related Quality of Life was measured by DLQI.

I think that would be enough: The effects of selected biologics and a small molecule on Health-Related Quality of Life (HRQoL) in adult plaque psoriasis patients – systematic review and meta-analysis". Quality of Life can be a keyword.

Methods

In "Search strategy":... papers of all languages. In "Eligibility criteria" ..... 2. non-English language...

Please clarify this.

Results: ".....the 43 studies, which met our inclusion criteria, the total of 25,898 individuals were evaluated by DLQI and....... (15,16,17,18,19) .......(20,21,22,23)" ......

Strengths and Limitations of the study (24, 26, 28, 35, 37, 38, 39 )

You should quote the references of sequential numbers by placing only a hyphen between the first and the last number and not mentioning all the numbers, as follows: 15-19 and 20-23 and 24, 26, 28, 35, 37-39

"As for ethnicity of the evaluated cases, most of the studies demonstrated differentiation in the patients’ race. Only 13/43 studies evaluated single-ethnic group (8 studies enrolled only Asian and 5 studies only Caucasian patients). Moreover, 11/43 papers suffered from the lack of data concerning the patients’ race."

Race and ethnicity have different meanings. You need to standardize the term. See also table 1

Conclusion

"This meta-analysis also compares biologics, a small molecule and placebo effect on DLQI in psoriatic patients."

You didn't say the conclusion about this small molecule

6. PLOS authors have the option to publish the peer review history of their article (what does this mean?). If published, this will include your full peer review and any attached files.

Reviewer #1: No

Reviewer #2: No

Reviewer #3: No

Reviewer #4: No

Reviewer #5: No

---

## [Author Response · Author response to Decision Letter 0]

23 Sep 2020

Answer to the Reviewer 1:

Thank you for your evaluation. I would like to answer your questions.

The present meta-analysis aims to assess the utility of biological agents and one small molecule inhibitor (Tocafitinib) in improving the quality of life of Psoriasis patients (as measured by the DLQI). Psoriasis can be a highly debilitating condition, and the advent of a number of new therapeutic options does call into need a systematic analysis of the different treatments. There is a recent network meta-analysis (2017) by Lopez et al. covering the same biological agents (infliximab, adalimumab, ixekizumab, secukinumab). Due to the network structure of the analysis, it was able to better compare the different biological agents. In addition, it had a somewhat wider coverage of outcomes, as it was not restricted solely to the DLQI as an outcome measure. There is also one recent meta-analysis (2019) by Tian et al. conducted specifically on the small molecule inhibitor Tocafitinib (assessing outcomes other than the DLQI; namely Physician global assessment response, PASI75 and PASI90) in which a significant improvement was seen.

In addition, I believe correcting the following points of methodology would probably improve the overall value of the manuscript and hopefully improve the chances of a successful publication:

The aforementioned previous Meta-Analyses:

1. Quantitative Evaluation of Biologic Therapy Options for Psoriasis: A Systematic Review and Network Meta-Analysis

2. Efficacy and Safety of Tofacitinib for the Treatment of Chronic Plaque Psoriasis: A Systematic Review and Meta-Analysis

Answer to the Reviewer 1: Thank you for your comments. Both meta-analyses you suggested have now been included in Discussion.

1. The authors state that 423 studies qualified for full-text screening; however, the PRISMA diagram shows that only 150 articles were assessed for eligibility.

Answer to the Reviewer 1: Thank you for your valuable remark.

We started our meta-analysis by reviewing the PRISMA guidelines derived from http://www.prisma-statement.org/ and https://www.ncbi.nlm.nih.gov/pmc/articles/PMC6461330/ and then two independent researchers selected 423 papers from different databases which met the PRISMA requirements. Later, after we had screened titles, keywords and abstracts of the selected 423 papers, it turned out that 273 papers (out of those originally chosen 423 papers) did not meet the PRISMA criteria because they lacked certain data (e.g. quality of life measured by means of adequate tools, non-psoriasis patients included in the study or papers, mixed type population of patients with different dermatologic or inflammatory diseases, a lack of analysis of drugs’ effect on the quality of life). Finally, we excluded the papers which were regarded as original at first but then turned out to be reviews or other types of articles, therefore, we were eventually left with 150 eligible papers.

2. The authors’ PRISMA diagram shows a total of 43 included studies; however, they then proceed to choose 17 papers for evaluation of select biologic agents without clarifying the rationale as to why these 17 studies were the chosen ones. This may well be justified due to the nature of the selected studies, but I believe it would be best for the authors to clearly outline the justification for their selection.

Answer to the Reviewer 1: Thank you for another observant remark. Obviously, we took a shortcut but following your advice we are including this “clear outline” in the present version of the manuscript.

The number 17 resulted from exclusion of 26 papers (out of originally chosen 43) because some of them did not include appropriate data for the analysis of the effect of selected biological agents and a small molecule on HRQL (e.g. changes in DLQI scoring prior to and after the commencement of therapy, comparison between placebo and drug effect), while others lacked biologic agents in their analysis (phototherapy, topical therapy, methotrexate or other non-biologic drugs were explored). 

However, we decided to include all of the 43 papers in our analysis in order to demonstrate the differences in DLQI scoring for different races. We believe that presented Table and Figure are helpful in noticing these differences.

3. The authors included a network diagram despite the fact that no network analysis was done. I think, for the sake of brevity, the diagram should probably be removed.

Answer to the Reviewer 1: Following the reviewer’s suggestion, for which we are grateful, the network diagram has been removed from the paper.

4. I do not believe the “citations” column in tables 1-3 serve to add much value, as the number of citations of included studies is not directly relevant to the review.

In addition, as the quality of writing greatly affects publication chances, and as the manuscript contains some linguistic errors, the paper would probably benefit from a grammatical revision to improve readability.

Answer to the Reviewer 1: Thank you very much for your valuable comments. Following the reviewer's comment, the citation column and number of citations in Tables 1-3 has been removed. Linguistic and grammatical errors have also been corrected.

Answer to the Reviewer 2

Thank you for your evaluation. 

This is a systematic review and meta-analysis in which the authors investigates the effects and efficiency of different biologics and many biological drugs on Health-Related Quality of Life (HRQoL) in adult plaque psoriasis patients measured by Dermatology Life Quality Index (DLQI). DLQI is commonly used tool to assess quality of life of patients with skin diseases. The authors found that Infliximab, adalimumab, ixekizumab, secukinumab and tofacitinib in adult plaque psoriatic patients improved HRQOL measured by DLQI. The manuscript is well-written and structured. It also provides a tool for the physician to choose the right biological agent to improve the quality of life in patients with psoriasis.

Answer to the Reviewer 2: Thank you for your evaluation. We appreciate your work.

Answer to the Reviewer 3

Thank you for your evaluation. I would like to answer your questions.

The authors performed a systematic review and meta-analysis to investigate the effect of infliximab, adalimumab, ixekizumab, secukinumab, and tofacitinib on Health-Related Quality of Life (HRQOL) measured by DLQI in adult plaque psoriatic patients.

Although the overall approach of this review is proper, I have few comments:

1. Title: abbreviations should be avoided. Also, replace (– systematic) with (: A systematic).

Answer to the Reviewer 3: Thank you for your remarks. The abbreviations in the title have been removed. Linguistic and grammatical errors have been corrected.

2. The short title is missing. 

Answer to the Reviewer 3: Thank you for your opinion. The short title has been added.

3. Abstract

a) Main Objectives: remove (Aim of the study)

b) Material and Methods

- Searched through XX, insert the last date searched.

- clearly outline the eligibility criteria

- Insert the software used for analysis.

c) The results are inadequately reported, please add data of the pooled effect estimates.

d) Keywords are missing.

Answer to the Reviewer 3: Thanks for your suggestions. The ,,Aim of the study’’ has been substituted with “Main Objectives”. The last search date has been added to the manuscript (October 31st, 2019). The eligibility criteria are written more clearly. The software used for analysis has been inserted (Meta XL software version 5.3.).

Pooled SMD which represent effect size (standardized mean difference) and 95% confidence interval (using a random-effects model) are indicated by diamond in each figure, however, according to reviewer's suggestion pooled effect estimates were included in table 2 and 3.

Keywords have been added to the manuscript. Keywords: psoriasis, dermatology life quality index, quality of life, biologic agents, tofacitinib, meta-analysis.

4. Introduction: place references at the end of sentences.

Answer to the Reviewer 3: References have been added at the end of the sentences. 

5. Methods

- The authors searched databases until October 2019. I would suggest updating the searching process for recently published relevant articles.

Answer to the Reviewer 3: Thank you for this comment. Two meta-analyses have been added to the discussion:

1. Quantitative Evaluation of Biologic Therapy Options for Psoriasis: A Systematic Review and Network Meta-Analysis

2. Efficacy and Safety of Tofacitinib for the Treatment of Chronic Plaque Psoriasis: A Systematic Review and Meta-Analysis

6. No data on how the risk of bias assessment was performed.

Answer to the Reviewer 3: We used ordinary methods (funnel plot analysis) and a quite new and reliable method (Luis Furuya-Kanamori index) to assess the risk of bias. The methodology and short description of the used method was described in materials and method section. We would like to introduce reviewer to this method and let us quote the Doi plots and Luis Furuya-Kanamori (LFK) index methods: “LFK generates the Doi plot and estimates the LFK index to detect and quantify asymmetry of study effects. The Doi plot replaces the conventional scatter (funnel) plot of precision versus effect with a folded normal quantile (Z-score) versus effect plot. The studies form the limbs of this plot, if there is asymmetry there will be unequal deviation of both limbs of the plot from the mid-point or more studies making up one limb compared to the other. In the absence of asymmetry, it would be expected that a perpendicular line to the X-axis from the tip of the Doi plot would divide the plot into two regions with similar areas. The LFK index quantifies the difference between these two regions in terms of their respective areas under the plot and the difference in the number of studies included in each limb. The closer the value of the LFK index to zero, the more symmetrical the Doi plot. LFK index values outside the interval between -1 and +1 are deemed consistent with asymmetry (i.e. publication bias).”

References:

https://econpapers.repec.org/software/bocbocode/s458762.htm

https://journals.lww.com/ijebh/Citation/2018/12000/A_new_improved_graphical_and_quantitative_method.3.aspx

6. Results

- Data on the screening process are incomplete and wrongly placed. Please move it to “Methods.”

- Data in Table 1 and Figure 2 are the same. Repeating data is not preferred; thus, I would recommend that one of them can be submitted as a supplementary file. The same is true for other tables and figures of the same outcome.

Answer to the Reviewer 3: Data on the verification process has been transferred to methods. Figures have been added to the supplementary file.

7. A completed PRISMA checklist is missing.

Answer to the Reviewer 3: A completed Prisma checklist has been added. 

8. Figures should be submitted in better resolutions.

Answer to the Reviewer 3: Thank you for your comment. We have provided figures in better resolution, now they are in 300 DPI.

9. Language: The entire manuscript needs extensive professional revision for grammatical errors and stylistic editing to improve the quality of English. For example,

- Page 1: (to assess the quality), not (to assess quality); (confirm the positive effect) not (confirm positive effect); (with an overall DLQI score) not (with overall DLQI score); (The random-effects model) not (Random effects model). (Scopus, ClinicalTrials.gov and manual searching) & (ixekizumab, secukinumab and tofacitinib), a comma should be placed before (and), etc.

Answer to the Reviewer 3: Linguistic and grammatical errors have been corrected.

Answer to the Reviewer 4

Thank you for your evaluation. I would like to answer your questions.

-There are some points to consider when you revise your manuscript.

The title of your study is too long. It would be better to make it short such as following;

(The effects of selected biologics and a small molecule on Health-Related Quality of Life: systematic review and meta-analysis.)

The affiliation sentence needs revision.

The reference need to be revised according to the journal guideline.

It also need English editing.

-Why did you include 43 studies while only 17 of them assessed the DQLI of patients on biologic treatments? based on your objective you should only include these 17 studies.

Answer to the Reviewer 4: 

-Thank you for your remarks. The manuscript’s title has been corrected, the affiliation sentences have also been revised. The references have been revised according to the journal guidelines. Linguistic and grammatical errors have been corrected.

Reviewer 4

Why did you include 43 studies while only 17 of them assessed the DQLI of patients on biologic treatments? based on your objective you should only include these 17 studies.

Answer to the Reviewer 4

The number 17 resulted from exclusion of 26 papers (out of originally chosen 43) because some of them did not include appropriate data for the analysis of the effect of selected biological agents and a small molecule on HRQL (e.g. changes in DLQI scoring prior to and after the commencement of therapy, comparison between placebo and drug effect), while others lacked biologic agents in their analysis (phototherapy, topical therapy, methotrexate or other non-biologic drugs were explored). 

However, we decided to include all of the 43 papers in our analysis in order to demonstrate the differences in DLQI scoring for different races. We believe that presented Table and Figure are helpful in noticing these differences.

Answer to the Reviewer 5

Title

I believe it is unnecessary to put in the title of the article that Health-Related Quality of Life was measured by DLQI. I think that would be enough: The effects of selected biologics and a small molecule on Health- Related Quality of Life (HRQqL) in adult plaque psoriasis patients- systematic reciew and meta-analysis’’. Quality of Life can by a keyword. 

Answer to the Reviewer 5: Thank you for your comments. The title of the manuscript has been changed to ,,The effects of selected biologics and a small molecule on Health-Related Quality of Life in adult plaque psoriasis patients - A systematic review and meta-analysis’’. “Quality of life” has been added as a keyword.

Methods:

In ,,search strategy’’ papers of all languages. In eligibility criteria 2. Non- English language. Please clarify this.

Answer to the Reviewer 5: Search strategy has been improved. The works were searched for, among papers which have been written in the English language. 

Results the 43 studies, which met our inclusion criteria, the total of 25,898 individuals were evaluated by DLQI and......(15,16,17,18,19).(20,21,22,23 ......Strengths and Limitations of the study (24, 26, 28, 35, 37, 38, 39 ).

You should quote the references of sequential numbers by placing only a hyphen between the first and the last number and not mentioning all the numbers, as follows: 15-19 and 20-23 and 24, 26, 28, 35, 37-39.

Answer to the Reviewer 5: Thank you for your remarks. We have made all corrections following the reviewer’s suggestions. 

"As for ethnicity of the evaluated cases, most of the studies demonstrated differentiation in the patients’ race. Only 13/43 studies evaluated single-ethnic group (8 studies enrolled only Asian and 5 studies only Caucasian patients). Moreover, 11/43 papers suffered from the lack of data concerning the patients’ race."

Race and ethnicity have different meanings. You need to standardize the term. See also table 1

Answer to the Reviewer 5: Thank you for your remarks. Everything has been corrected according reviewer’s suggestions. 

Conclusion:

"This meta-analysis also compares biologics, a small molecule and placebo effect on DLQI in psoriatic patients."

You didn't say the conclusion about this small molecule.

Answer to the Reviewer 5: 

Thank you for your comment. “A small molecule” has been added in Conclusion.

---

## [Decision Letter · Decision Letter 1]

19 Oct 2020

The effects of selected biologics and a small molecule on Health-Related Quality of Life in adult plaque psoriasis patients: A systematic review and meta-analysis.

PONE-D-20-16022R1

Dear Dr. Karpinska-Mirecka,

We’re pleased to inform you that your manuscript has been judged scientifically suitable for publication and will be formally accepted for publication once it meets all outstanding technical requirements.

Kind regards,

Ahmed Negida, MD

Academic Editor

PLOS ONE

Additional Editor Comments (optional):

Reviewers' comments:

Reviewer's Responses to Questions

**Comments to the Author**

1. If the authors have adequately addressed your comments raised in a previous round of review and you feel that this manuscript is now acceptable for publication, you may indicate that here to bypass the “Comments to the Author” section, enter your conflict of interest statement in the “Confidential to Editor” section, and submit your "Accept" recommendation.

Reviewer #3: All comments have been addressed

Reviewer #4: All comments have been addressed

Reviewer #5: All comments have been addressed

2. Is the manuscript technically sound, and do the data support the conclusions?

Reviewer #3: Yes

Reviewer #4: Yes

Reviewer #5: (No Response)

3. Has the statistical analysis been performed appropriately and rigorously? 

Reviewer #3: Yes

Reviewer #4: Yes

Reviewer #5: (No Response)

4. Have the authors made all data underlying the findings in their manuscript fully available?

Reviewer #3: Yes

Reviewer #4: Yes

Reviewer #5: (No Response)

5. Is the manuscript presented in an intelligible fashion and written in standard English?

Reviewer #3: Yes

Reviewer #4: Yes

Reviewer #5: (No Response)

6. Review Comments to the Author

Reviewer #3: The authors have addressed all my comments/suggestions.

Reviewer #4: None

Reviewer #5: (No Response)

7. PLOS authors have the option to publish the peer review history of their article (what does this mean?). If published, this will include your full peer review and any attached files.

Reviewer #3: No

Reviewer #4: No

Reviewer #5: **Yes: **Marilda Aparecida Milanez Morgado de Abreu

---

## [Editor Report · Acceptance letter]

13 Nov 2020

PONE-D-20-16022R1 

The effects of selected biologics and a small molecule on Health-Related Quality of Life in adult plaque psoriasis patients: A systematic review and meta-analysis 

Dear Dr. Karpińska-Mirecka:

I'm pleased to inform you that your manuscript has been deemed suitable for publication in PLOS ONE. Congratulations! Your manuscript is now with our production department. 

Kind regards, 

on behalf of

Dr. Ahmed Negida 

Academic Editor

PLOS ONE